# Comparative Pharmacokinetics and Bioequivalence of Pour-On Ivermectin Formulations in Korean Hanwoo Cattle

**DOI:** 10.3390/antibiotics13010003

**Published:** 2023-12-19

**Authors:** Suyoung Kim, HyunYoung Chae, Eon-Bee Lee, Gayeong Lee, Seung-Chun Park, Jeongwoo Kang

**Affiliations:** 1Laboratory of Veterinary Pharmacokinetics and Pharmacodynamics, Institute for Veterinary Biomedical Science, College of Veterinary Medicine, Kyungpook National University, Daegu 41566, Republic of Korea; yan111@knu.ac.kr (S.K.); eonbee@knu.ac.kr (E.-B.L.); yeong1129@knu.ac.kr (G.L.); 2Animal Disease Diagnosis Division, Animal and Plant Quarantine Agency (APQA), Ministry of Agriculture, Food and Rural Affairs, 177, Hyeoksin 8-ro, Gimcheon-si 39660, Republic of Korea; iichy33ii@korea.kr; 3Laboratory of Veterinary Physiology, College of Veterinary Medicine, Kyungpook National University, Daegu 41566, Republic of Korea

**Keywords:** ivermectin, bioequivalence, pharmacokinetics, Korean native cattle (Hanwoo)

## Abstract

This study aimed to conduct a bioequivalence study of applying three pour-on ivermectin formulations at a dose of 1 mg/kg on the back of Korean native beef cattle (Hanwoo). To conduct bioequivalence testing, the pharmacokinetics of three groups (control Innovator, test Generic A, and test Generic B) of five clinically healthy Korean Hanwoo cattle (average weight 500 kg) were studied. After topical application to the skin, blood samples were drawn at the indicated times. These blood samples were analyzed using liquid chromatography–tandem mass spectrometry (LC-MS/MS). The time required to reach the maximum concentration (T_max_), the maximum concentration (C_max_), and the area under the curve (AUC_last_) of each pharmacokinetic parameter were compared for bioequivalence. The results showed that the control had a T_max_ of 41 ± 1.24 h, a C_max_ of 0.11 ± 0.01 μg/mL, and an AUC_last_ of 9.33 ± 0 h*μg/mL). The comparator Generic A had a T_max_ of 40 ± 1.14 h, a C_max_ of 0.10 ± 0.01 (μg/mL, and an AUC_last_ of 9.41 ± 0.57 h*μg/mL, while Generic B had a T_max_ of 40 ± 2.21 h, a C_max_ of 0.10 ± 0.01 μg/mL, and an AUC_last_ of 9 h*μg/mL. The values of the bioequivalence indicators C_max_, T_max_, and AUC were all within the range of 80% to 120%, confirming that all three tested formulations were bioequivalent. In conclusion, the study showed that the two generic products were bioequivalent to the original product in Hanwoo cattle.

## 1. Introduction

Ivermectin is a well-known drug that has received approval from both the US Food and Drug Administration and the World Health Organization for its use as an antiparasitic medication [1]. Ivermectin is a macrocyclic lactone with high endectocide efficacy against both internal and external parasites. It also exhibits long-lasting antiparasitic activity [2]. It is commonly employed in low- and middle-income nations to treat worm infections [3]. Over its more than 25 years of use, researchers have discovered that it possesses antiviral and anti-inflammatory properties, thereby expanding its potential applications in medicine [4]. Ivermectin is readily available and cost-effective in many regions worldwide [5]. Among various anti-ectoparasitic drugs, it has been shown to be a beneficial medicine for managing cattle on pastures and acts as an effective tick repellent [6].

The pharmacokinetics of ivermectin have been reported in various animal species, including cattle, sheep, goats, pigs, horses, and dogs. Ali & Hennessy (1996) investigated the pharmacokinetic disposition and efficacy of ivermectin in sheep after administration of a feed mixture that contained it [7]. Echeverria et al. (1997) reported the pharmacokinetics of ivermectin after intravenous and subcutaneous administration in cattle [8]. Gonzalez et al. (2006) conducted a study on the pharmacokinetics of a new formulation of ivermectin in goats [9]. Craven et al. (2001) reported the pharmacokinetics of moxidectin and ivermectin after intravenous injection in pigs with different body compositions [10]. Gokbulut et al. (2001) investigated the plasma pharmacokinetics and fecal excretion of ivermectin, doramectin, and moxidectin after oral administration in horses [11]. Based on these reports, pharmacokinetic models can be used to indicate one or two compartments, depending on the route of administration and the formulation used. In addition, Lifschitz et al. (2004) performed a pharmacokinetic evaluation of four generic formulations in calves and found various pharmacokinetic parameters, including C_max_, T_max_, and AUC, in the aforementioned animal species [12]. These results suggest that the pharmacokinetic profile of ivermectin may be influenced by factors such as the body fat composition of the animal and the composition of the ivermectin formulation.

Hanwoo cattle, which are small-bodied, gradually maturing, and adaptable animals known for generating superbly flavored and marbled meat [13], are currently raised as beef cattle in South Korea and are an economic mainstay in the livestock industry [14]. Several ivermectin formulations have been used in veterinary drugs without any reports on its pharmacokinetics in Korean cattle [6]. For this reason, the availability of generic products is necessary, and establishing a bioequivalence assessment in Hanwoo cattle is important. Bioequivalence studies play a pivotal role in determining the therapeutic interchangeability of different medicinal products and have been conducted in various animal species [15,16].

Hanwoo cattle are primarily raised in South Korea, where their meat is highly prized. Therefore, drugs applied to these cattle are based on beef cattle. However, the unique physiological and metabolic characteristics of Hanwoo cattle have not been well studied. Unlike Holstein cattle, Hanwoo is a breed of small cattle that is native to Korea. Previously, they were used as raw livestock, although this has all but disappeared, and they are now primarily raised as a meat source. They have brown fur, while both males and females have horns. Their maternal qualities are good, yet they produce relatively little milk—less than 400 L over a 170-day lactation period [17]. Considering this knowledge gap, the present study aimed to explore bioequivalence based on the pharmacokinetic results for commercially available topical ivermectin formulations in Korean Hanwoo cattle. Potential differences in drug absorption, metabolism, distribution, and elimination of these products may play a key role in the decision-making process of veterinarians and livestock producers regarding drug application, dosage, and treatment [18].

For this purpose, this study was performed to obtain pharmacokinetic data for the original patented ivermectin product and two generic products (three top-selling ivermectin-containing pour-on medications) that are currently available in the Korean market for Hanwoo cattle. Each product was applied at the same dose using the same method of administration. Subsequently, using the obtained pharmacokinetic parameters, the bioequivalence of these existing three ivermectin products was determined in Korean Hanwoo cattle.

## 2. Results

### 2.1. Validation of Ivermectin Quantification Methods

The peak for the ivermectin extracted from the standard solution was observed in the chromatogram at approximately 6.3 min, as shown in Figure 1. Using the optimized settings, the LC–MS/MS system showed an improved symmetric peak for ivermectin in both the standard solution and plasma matrix.

Ivermectin was extracted quickly, efficiently, and simply using the optimized analytical technique. The LC–MS/MS analysis did not require derivatization, which is applied to reduce the analysis time. To assess the validity of the ivermectin concentration assay in plasma, its linearity, selectivity, accuracy, and precision were evaluated. These results are summarized in Table 1.

Linearity was assessed using a calibration curve derived from three different concentrations (25, 50, and 100 ng/mL) of a standard solution. As shown in Table 1, the analytical method demonstrated excellent linearity with the calibration curve, showing an r^2^ value greater than 0.99. The recovery and precision of ivermectin were evaluated at concentrations of 25, 50, and 100 ng/mL, with three replicates analyzed at each concentration level. The mean recovery of ivermectin was calculated at 98%, and the coefficient of variation percentage (CV) was less than 10%. The sensitivity of the method was determined by assessing the limits of detection (LOD) and limits of quantification (LOQ) values using the signal-to-noise ratio measurements. The analyte concentrations that corresponded to signal-to-noise ratio values of 3 and 10 were defined as the LOD and LOQ, respectively. Thus, the LOD and LOQ values for ivermectin were 3 and 10 ng/mL, respectively.

### 2.2. Pharmacokinetic Analysis and Bioequivalence

To obtain the pharmacokinetic parameters for Korean Hanwoo cattle, the plasma ivermectin concentration was measured over time after topical administration of three products (Innovator, Generic A, and Generic B) at a dose of 1 mg/kg, and the results are shown in Figure 2.

As shown in Figure 3, all three products showed the same time–concentration changes, and the pharmacokinetics were calculated by non-compartmental analysis (NCA), as shown in Table 2.

Table 2 provides the pharmacokinetic parameters related to the pour-on applications of ivermectin. The Innovator, Generic A, and Generic B products showed ivermectin maximum concentrations (C_max_) of 0.11, 0.10, and 0.10 μg/mL, respectively. The area under the last observable plasma concentration-time curve (AUC_last_) following pour-on administration was 9.75 h*μg/mL, 9.33 h*μg/mL, and 9.41 h*μg/mL for the Innovator, Generic A, and Generic B products, respectively. The time to reach maximum concentration (T_max_) was 40 h for each of the Innovator, Generic A, and Generic B products. To determine the differences in bioequivalence for the three formulations, the C_max_ and AUC values are shown in Figure 3.

Pharmacokinetic analyses of the three formulations in this study showed that the Innovator, Generic A, and Generic B C_max_ and AUC values were within the range of 80–120%, in the standard outlined in Article 17 (Evaluation) of the Regulations on Drug Equivalence Test Standards, previously established by the Ministry of Food and Drug Safety [19]. Thus, it was confirmed that all three products are bioequivalent.

## 3. Discussion

Ivermectin, a broad-spectrum parasiticide, is well-suited for sheep, and its pour-on formulation can efficiently and conveniently treat ectoparasitic infection [20]. Ivermectin has particularly antiparasitic effects against *Ostertagia ostertagi* and *Cooperia oncophora*. Moreover, ivermectin has antibacterial activity against *Chlamydia trachomatis* and *Mycobacteria* spp., as well as antiviral and anti-inflammatory properties [21]. For this reason, the pharmacokinetics and pharmacodynamics of ivermectin have been studied in several animals [22]. Recently, Choi et al. (2019) studied the efficacy of ivermectin against *Theieria orientalis* infection in grazing cattle (Holstein cattle) [6]. From their study, ivermectin offered protection against *Theieria orientalis* and RBC hemolysis in cattle grazing in Korea, although no pharmacokinetic information was presented. Currently, ivermectin preparations are being used in Korean cattle, yet no information on the efficacy is available. Three ivermectin formulations are widely used in Korea as a topical pour-on method for ectoparasite control in native cattle, although there is no information on its pharmacokinetics.

An important procedure required to perform pharmacokinetic studies on ivermectin is to determine the low-level concentrations remaining in the blood after its administration to cattle. Analysis of ivermectin concentrations in the blood was performed by HPLC. When performing HPLC, fluorescence analysis was used after the derivatization of ivermectin to measure the low concentrations of the drug in the blood. Kitzman et al. (2006) used HPLC to analyze ivermectin levels in human plasma after administration, following the fluorescence induction of ivermectin by trifluoroacetic anhydride (TFAA) and N-methylimidazole (NMI) [23]. The results showed that the limit of quantitation (LOQ) for ivermectin in human plasma was 0.2 ng/mL, which was more than five times the baseline noise observed for the retention time of ivermectin. The coefficient of variation (n = 6) of the measured concentration at LOQ was 6.1%, and the deviation of the measured concentration from the mean and nominal value was 4.3%. However, this method is subject to losses occurring during the fluorescence induction process, so the use of liquid chromatography–tandem mass spectrometry (LC-MS/MS), which extracts and analyzes the drug directly from plasma, may improve the accuracy of analyzing ivermectin in plasma. The analytical method is deemed linear when the correlation coefficient (r^2^) is greater than 0.9900 [24]. As per the criteria for developing and validating analytical methods, the precision (expressed as the coefficient of variation % CV) of the optimized LC–MS/MS method was within the accepted limits [25]. The meticulous methodology adhering to the standards set by the Ministry of Food and Drug Safety serves as a guide for any subsequent investigations in this field [26].

Most reported that animal pharmacokinetics are based on intravenous, intramuscular, subcutaneous, and oral administrations. As a result, pharmacokinetic parameters following pour-on administration are rarely reported. In this study, we used the LC–MS/MS system, which holds the potential to provide quicker and more precise outcomes. The drug concentration curve in plasma after administering ivermectin to cattle at a dose of 1 mg/kg·bw is shown in Figure 2. Interestingly, the time–concentration curves for ivermectin in the blood after administering the three products tested were almost identical. Although bioequivalence studies for ivermectin exist in several animals, including cattle [27], there are no reports of such studies in Korean Hanwoo cattle. In this study, pharmacokinetics analysis was performed for the first time, and a bioequivalence assessment of three commercially available formulations was performed, ultimately confirming bioequivalence. Ivermectin has been the subject of many pharmacokinetic studies since its introduction in 1981 [28]. Depending on the species, oral, intramuscular (IM), subcutaneous (SC), or topical administration has been used.. Blood was collected after topical administration at the specified time and analyzed by LC–MS/MS to determine the plasma concentration, as shown in Figure 2. In cattle, the behavior of the drug in the body after intravenous and intramuscular injections of ivermectin showed a two-compartment model and a one-compartment model, respectively [29,30]. However, Lanusse et al. (1997) analyzed the two-compartment model [31]. Gayrard et al. (1999) analyzed the one-compartment model behavior in cattle after pour-on administration, which was consistent with the one-compartment behavior shown in this study [32]. Ivermectin has a C_max_ of 0.022 μg/mL after intramuscular injection [29]. Subcutaneous administration, the most commonly used route of administration, reported a 6-fold difference in C_max_ from a low of 0.022 μg/mL to a high of 0.133.2 μg/mL for the same formulation and different formulations [18,29,30]. This was thought to be due to differences in the HPLC methods, differences in the formulations, and the method of administration. Gayrard et al. (1999) reported a C_max_ of 0.012 μg/mL and a T_max_ of 81.6 h after administering ivermectin using the same topical application [31]. Compared to the pharmacokinetic parameters obtained, the C_max_ was 10 times higher, and the T_max_ was 2 times faster in this study. These differences were thought to be due to differences in the HPLC and LC/MS analytical methods, in addition to the loss of the drug in cattle experiments due to the habit of licking. However, this study was conducted using individual cages to prevent this. Therefore, in this study, a dose of 1 mg/kg was applied, which is twice as high as the previously used dose, considering the route of administration and previous C_max_ results.

Bioequivalence studies are commonly performed following the guidelines proposed by the U.S. Food and Drug Administration for human pharmacokinetics studies [32]. A notable aspect of creativity emerges through the comparison of pharmacokinetics between the original Innovator product and two generic versions. Table 2 provides the pharmacokinetic parameters related to the pour-on applications of ivermectin. Several formulations of ivermectin were made and tested for bioequivalence after subcutaneous injection [15,18,28,29]. The results showed significant differences between the formulations. Lo et al. (1985) prepared an aqueous vehicle: aqueous–glycerol-formal vehicle (50:50, *v*/*v*) and a propylene–glycol:glycerol-formed vehicle (60:40, *v*/*v*) and compared them [18]. The results showed that the aqueous vehicle formulation tended to be the most bioavailable, with a bioavailability of 55%; however, the propylene–glycol:glycerol-formed vehicle formulation showed a steadily higher C_max_ of 0.046 μg/mL [12,15,29]. In this study, no statistical differences were seen when comparing the pharmacokinetic parameters of the three formulations; however, the Innovator and Generic B formulations tended to exhibit a longer T_1/2_ and higher Vz/F than those observed for the Generic A formulation. These differences were likely since the Innovator and Generic B formulations contained the same concentration of propylene glycol, while the Generic A formulation excluded this ingredient. From these results, it can be inferred that propylene glycol may affect the critical T_1/2_ and Vz/F when applied topically (pour-on). All three formulations used in the present study were formulated in various fat-soluble solvents with or without propylene glycol. After topical application of these formulations at 1 mg/kg, all of them showed C_max_ of 0.1–0.11 μg/mL, T_max_ of 40 h, and AUC_last_ of 9.33–9.75 h*μg/mL, while also being found to be bioequivalent. However, the effect of propylene glycol is not statistically significant in the study. Despite the result, additional studies on the formulations would be an urgent and important topic for further research.

In summary, the pharmacokinetic characteristics of three commercially available ivermectin products were determined by administering a 1 mg/kg dose to Hanwoo cattle using a pour-on method. The study revealed that the maximum concentration (C_max_), time to reach maximum concentration (T_max_), and area under the curve (AUC) of these three products are bioequivalent, falling within the 80–120% range. The research also emphasizes the importance of comparing the original patented ivermectin with two generic versions. Verifying their bioequivalence suggests that these less expensive generic options can be used reliably, offering cost savings without sacrificing treatment efficacy.

## 4. Materials and Methods

### 4.1. Chemicals, Reagents, and Media

Ivermectin standard was provided by Sigma Aldrich (St. Louis, MO, USA). Acetonirile was purchased from Merck (Darmstadt, Germany). Formic acid was supplied by Fisher Scientific (Pittsburgh, PA, USA). All solvents used in the analysis were LC–MS grade. Purified water was obtained using a Milli-Q system (Millipore, Bedford, MA, USA).Innovator (Formulation A, Group A): SY Himecin (Samyang Anipharm): Ingredients and amounts (out of 1 L of the main preparation): Ivermectin (5 g), isopropanol (794.9 mL), propylene glycol (8 g), isopropyl myristate (160 g), oleyl alcohol (32 mL), butylated hydroxytoluene (0.1 g), food blue no. 1 (appropriate amount). Usage and capacity: Apply 0.1 mL of the main product (0.5 mg as ivermectin) per kg of body weight as a single dermal application along the midline of the back.Generic A (Formulation B, Group B): Gmectin-Pour On (GREEN CROSS Veterinary Products): Ingredients and amounts (out of 1 mL of the main preparation): Ivermectin (5 mg), triethanolamine (0.5 mg), food blue no. 1(0.01 mg), diethylene glycol monoethyl ether, isopropyl myristate, isopropanol (appropriate amount). Usage and capacity: Dermal application of 1 mL of the product per 10 kg of body weight.Generic B (Formulation C, Group C): IMEC-Pouron (SF company): Ingredients and amounts (out of 1 L of the main preparation): Ivermectin (5 g), propylene glycol (8 g), isopropanol, isopropyl myristate, food blue no. 1 (appropriate amount). Usage and capacity: Apply 0.1 mL of vehicle (0.5 mg as ivermectin) per kg of body weight as a single dermal application along the midline of the back.

### 4.2. Animal Experimental Procedure and Treatments

The Hanwoo experiment was conducted at the Korea Animal Testing Center (KULF), which specializes in preclinical and clinical trials. The animals were housed individually in an environment with a 12 h light/dark cycle. The ambient temperature was maintained at 25 to 28 °C. The animals had free access to food and water without restriction during the study. Fifteen male Hanwoo cattle weighing 500 ± 30 kg and aged between 18 and 20 months were randomly divided into 3 groups of 5 animals each and treated as described below.

Innovator (Formulation A, Group A): This group was treated with the ivermectin test formulation at a dose of 1 mg/kg per body weight. The pour-on solution was meticulously applied using a 1 L measuring cup. The application was along the topline of the cattle, forming a continuous slender strip stretching from the withers to the tail, as visually represented in Figure 4. This particular preparation was designated as the reference (Innovator) product for the study. Animals in the Generic A and B groups were administered the ivermectin test formulations using an identical methodology and dosage to those used in the Innovator group. For the purposes of this bioequivalence trial, the formulations given to the Generic A and B groups were categorized as the test products.

To mitigate any cross-contamination or interference, specifically the potential for animals to lick the formulations off of one another post-treatment, all cattle were securely housed in individual enclosures. The animal study was approved by the Institutional Animal Care and Use Committee (IACUC) of the Ministry of Agriculture, Food and Rural Affairs, Korea (approval number: 2021-470) and was conducted in accordance with animal testing guidelines.

### 4.3. Collection and Processing of Blood Samples

Ivermectin was applied to each subject, and blood was collected and monitored for 12 days, followed by 7 days for abnormalities. Blood samples of 3 mL were taken from the jugular veins of each subject at intervals of 8, 16, 24, 32, 40, 48, 56, 64, 72, 80, 88, 96, 104, 112, 120, 128, 136, 144, 152, 160, 168, 176, 184, 192, 216, 240, 264, and 288 h using vacutainer heparin tubes (Becton Dickinson and Company, Franklin Lakes, NJ, USA). The blood samples were centrifuged at 2000× *g* for 10 min at 4 °C to obtain plasma samples. A total of 1.5 mL of 0.1% formic acid in acetonitrile was added to 0.5 mL of plasma to collect plasma proteins. The mixtures were mixed for 20 min before being centrifuged at 5000× *g* for 30 min. Nitrogen was used to evaporate the supernatant fluid at 50 °C until the volume was reduced to 500 µL, after which the samples were kept in a 70 °C freezer.

### 4.4. Liquid Chromatography–Tandem Mass Spectrometry (LC–MS/MS) Analysis

The ivermectin serum concentration was assayed using a Shimadzu LC-MS 8045 triple–quadrupole mass spectrometer outfitted with a Nexera X2 ultra-high-performance liquid chromatography system (Shimadzu, Kyoto, Japan) and connected to an electrospray ionization (ESI) interface. Chromatographic separation was performed using an Xbride BEH C18 column measuring 2.1 × 100 mm with a particle size of 2.5 μm (Waters, Milford, KS, USA) at 40 °C and a flow rate of 0.4 mL/min, with an injection volume of 2 μL. A gradient program for the mobile phase was set as follows: mobile phase A of 0.1% formic acid in distilled water and mobile phase B of 0.1% formic acid in acetonitrile. A gradient program was used with a total run time of 10.00 min: 0.00–1.00 min at 10% B; 1.01–5.00 min at 95% B; 5.01–7.00 min at 95% B; 7.01–10.00 min at 10% B. Ionization was performed using an electrospray ionization source in positive mode. The MS/MS condition parameters were optimized as follows: interface temperature of 150 °C, heating gas flow of 10 L/min, DL temperature of 250 °C, heating block temperature of 400 °C, drying gas flow of 10 L/min, and nebulizing gas flow of 3 L/min. To quantify ivermectin, multiple reaction monitoring was performed. Chromatographic separation was performed using an Xbride BEH C18 column measuring 2.1 × 100 mm with a particle size of 2.5 μm (Waters, Milford, KS, USA) at 40 °C and a flow rate of 0.4 mL/min, with an injection volume of 2 μL.

### 4.5. Standard Solution Preparation

Ivermectin was accurately weighed and transferred to a volumetric flask, where it was dissolved in methanol (MeOH) to yield a stock solution with a concentration of 1.0 mg/mL. This solution was stored in a polypropylene tube, kept at a temperature of 4 °C, and sealed in bottles until needed. A working solution was prepared by diluting the stock solution with more methanol to attain the desired concentration.

### 4.6. Validation of Ivermectin Quantification Methods

The specificity of the ivermectin quantification method was evaluated using a standard ivermectin solution, a known amount of ivermectin in bovine plasma, and untreated bovine plasma. This was to determine if any interference from the matrix occurred at the ivermectin retention time. Ivermectin was dissolved in 0.1% aqueous formic acid to produce a 1 mg/mL stock solution. This solution was further diluted to generate various standard ivermectin solutions. To create ivermectin-spiked plasma samples at concentrations of 25, 50, and 100 ng/mL, different volumes of the ivermectin solutions were added to the plasma of untreated cows. All samples were filtered using a 0.2 µm polyvinylidene fluoride (PVDF) syringe filter for LC–MS/MS analysis. Each sample was analyzed three times, and the ivermectin concentrations were measured by LC–MS/MS following sample injection.

### 4.7. Pharmacokinetic Analysis and Bioequivalence

The concentration–time profiles acquired from the plasma of individual animals were subjected to analysis using the WinNonlin software (Version 6.1), developed by Statistical Consultants Inc. (Lexington, KY, USA). Non-compartmental analysis (NCA) was employed to assess the pharmacokinetic parameters of each animal. The use of NCA allowed the determination of key pharmacokinetic parameters. In addition, the analysis included the determination of each maximum plasmatic ivermectin concentration (C_max_) and the corresponding time required to reach the peak value (T_max_). The terminal-phase disposition rate constant (λ) was estimated through a least-squares linear regression of the natural log-transformed ivermectin concentrations over time. Subsequently, the half-life (T_1/2_) was calculated using the formula: t1/2 = 0.693/λ.

### 4.8. Statistical Analysis

The results are represented as the mean ± standard deviation (SD) derived from three repeated examinations. Statistical analyses were conducted using the F-test and one-way ANOVA. A *p*-value of less than 0.05 was considered to indicate statistical significance. In the assessment of bioequivalence between formulations, a 90% confidence interval approach using ANOVA on logarithmically transformed data was adopted. This analysis accounts for the sequence, subject variability, period, and formulation as fixed effects, while non-parametric methods are deemed unsuitable for this evaluation [25].

## 5. Conclusions

This study characterized the pharmacokinetics of a patented brand of pour-on ivermectin (Innovator), an antiparasitic, after application to Hanwoo cattle. In addition, two generic formulations (Generic A and Generic B) of ivermectin were prepared and tested for bioequivalence in order to expand its application in Hanwoo cattle. As a result, the AUC and C_max_ values of all three formulations tested in beef cattle showed the same levels, confirming their bioequivalence according to the ‘Guideline for Bioequivalence Testing of Veterinary Drugs’. In conclusion, the pharmacokinetic parameters, C_max_, AUC, and T_max_ of the three products were compared after applying the same dosages to the skin of Korean beef cattle, and they were all in the range of 80–120%, confirming that all three products are bioequivalent.

## Figures and Tables

**Figure 1 antibiotics-13-00003-f001:**
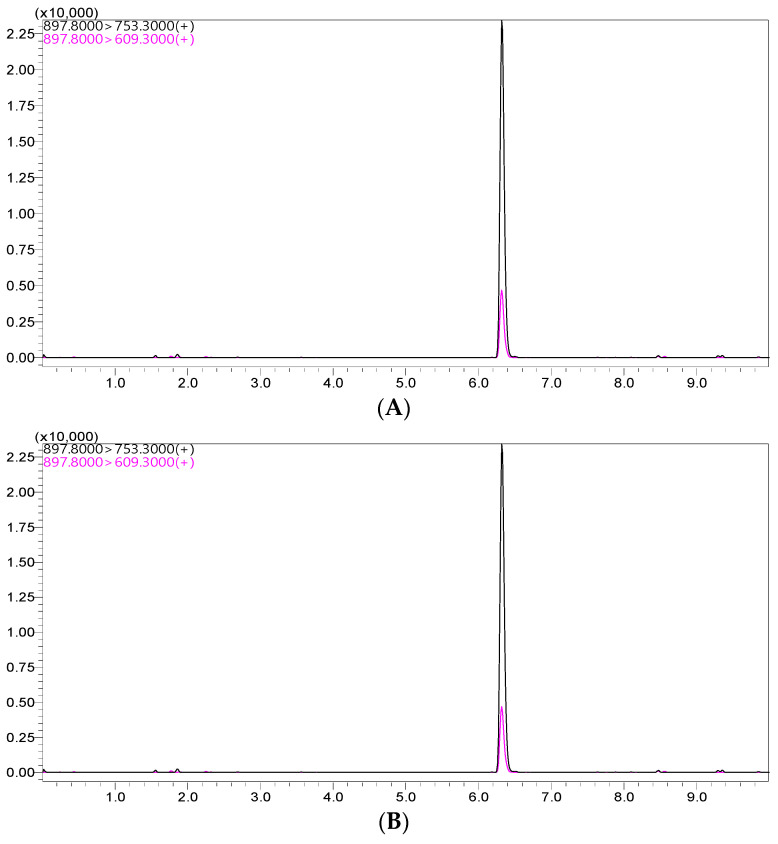
Representative LC–MS/MS chromatograms for ivermectin from the spiked sample (**A**) and standard solutions (**B**).

**Figure 2 antibiotics-13-00003-f002:**
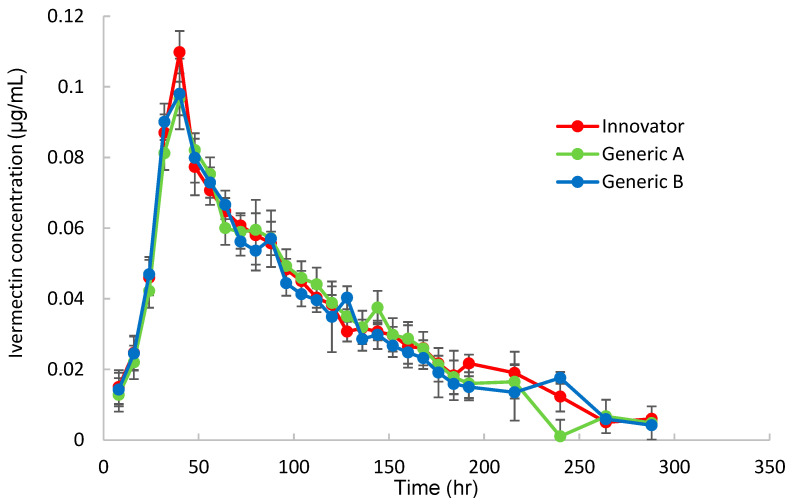
Concentration(μg/mL)–time profiles for ivermectin in plasma following pour-on applying the control Innovator or test Generic A and Generic B ivermectin formulations to cattle.

**Figure 3 antibiotics-13-00003-f003:**
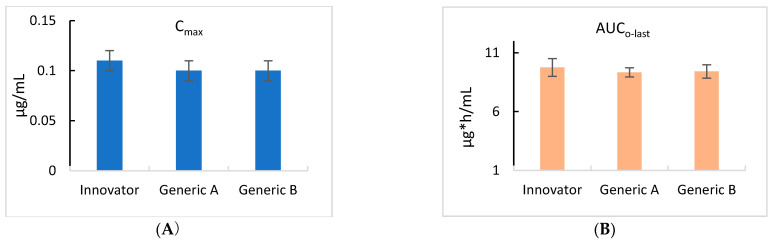
Comparison of the maximum concentration (Cmax) (**A**) and the area under the curve to the last measurable concentration (AUClast) (**B**) in cattle for the control Innovator, Generic A, and Generic B ivermectin formulations. The values for Cmax and AUClast for each of the three products were identical and showed no statistically significant variations.

**Figure 4 antibiotics-13-00003-f004:**
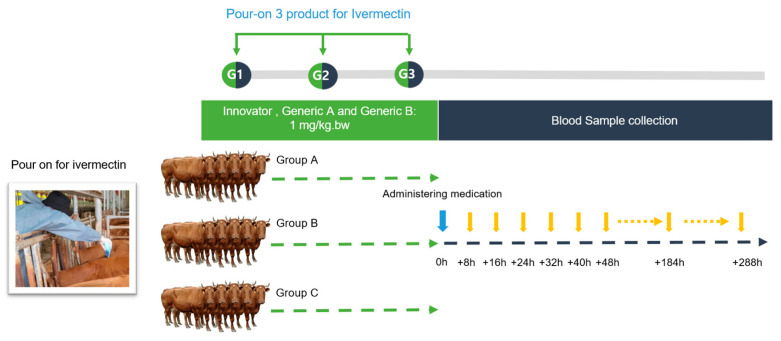
Experimental design to determine how ivermectin formulations work on Korean Hanwoo cattle. Fifteen Korean beef cattle were randomly divided into three groups: Innovator (group A), Generic A (group B), and Generic B (group C). Ivermectin was applied in a continuous strip along the midline of the upper back to the tip of the tail. After application, the animals were individually housed to prevent the possibility of licking each other.

**Table 1 antibiotics-13-00003-t001:** Validation of an LC–MS/MS method for quantifying ivermectin in the plasma of cattle.

Substance	RT (min)	Linearity (R²)	Average Recovery (%), (n = 5)	Coefficient of Variation (CV, %)	LOD (ng/mL)	LOQ (ng/mL)
Ivermectin	6.3	0.99	98	4.8	3	10

LC–MS/MS, liquid chromatography–tandem mass spectrometry; RT, retention time; LOD, limit of detection; LOQ, limit of quantitation.

**Table 2 antibiotics-13-00003-t002:** Main pharmacokinetic properties of patented and generic ivermectin formulations in cattle after pour-on administration at 1 mg/kg.bw.

Parameters	Innovator	Generic A	Generic B
T_1/2_ (h)	65.45	51.64	62.55
T_max_ (h)	40.00	40.00	40.00
C_max_ (μg/mL)	0.11	0.10	0.10
AUC_last_ (hr*μg/mL)	9.75	9.33	9.41
Vz/F (mL/kg)	9156.71	7692.57	9221.14
Cl/F (mL/h/kg)	96.98	103.25	102.18
MRT (h)	100.14	96.72	98.96

T_1/2_ (h), half-life; T_max_, time of maximum concentration; C_max_, maximum concentration after administration; AUC_last_, area under the concentration vs. time curve from the first observed to last measurable concentration; Vz/F, apparent volume of distribution; Cl/F, apparent clearance; MRT, average measure of time.

## Data Availability

All data generated for this study are contained within the article.

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
