# Peer review of "Comparative Pharmacokinetics and Bioequivalence of Pour-On Ivermectin Formulations in Korean Hanwoo Cattle"

_antibiotics, 2023, doi:10.3390/antibiotics13010003_

Round 1

Reviewer 1 Report

Comments and Suggestions for Authors

Innovator, Generic A and Generic B  are three veterinary medicinal products authorised and marketed, for which the generics were subject to bioequivalence studies facing the innovator during the authorisation procedure, likely the other animal breed (e.g. Holstein) has been applied. In the present manuscript, the authors consider it crucial to repeat the bioequivalence approach by Korean Hanwoo cattle since the metabolic characteristics of this cattle breed have not been well-studied, and bioequivalence studies results may be essential regarding the ivermectin application, dosage, and treatment. However, this project is not complete. Several critical parameters are missing. Despite the demonstrated bioequivalence, the efficacy of ivermectin by Korean Hanwoo cattle should be established or discussed. Moreover, the faecal ivermectin detection results are not described as was pointed out in the text, line 410: the concentration-time profiles acquired from plasma and faecal samples from individual animals were subjected to analysis using the WinNonlin software. Why were the animals treated with a dose of 1 mg/kg of body weight?

Manuscript:

1)      The abstract is too extended and should be more concise for easy reading.

2)      Introduction: …there are few pharmacokinetic and clinical studies on the application of veterinary drugs in Korean cattle, particularly ivermectin…Line 78-82 The reason for the present bioequivalence study (innovator versus generics) (?) Please discuss.

3)      Results: Concentration(ppm)–time (?) – line: 145; pour-on administration at 1000 g/kg.bw to cattle – line 153; Drug Equivalence Test Standards established by the Ministry of Food and Drug Safety – please reference – Line 189-190

4)      Discussion:

a)       Ivermectin is widely used in Korea as a topical pour-on method in Korean beef cattle for ectoparasite control, but there is no information on its pharmacokinetics- (lines 198-200). How were established the MRLs for ivermectin in the innovator?

b)      The discussion should be more concise, considering the results and possible constraints  related to this study (e.g. Korean Hanwoo features concerning other breed cattle)

Author Response

Point 1 : Innovator, Generic A and Generic B  are three veterinary medicinal products authorised and marketed, for which the generics were subject to bioequivalence studies facing the innovator during the authorisation procedure, likely the other animal breed (e.g. Holstein) has been applied. In the present manuscript, the authors consider it crucial to repeat the bioequivalence approach by Korean Hanwoo cattle since the metabolic characteristics of this cattle breed have not been well-studied, and bioequivalence studies results may be essential regarding the ivermectin application, dosage, and treatment.

Response 1 :

Thank you for your valuable comments. We acknowledge that Innovator, Generic A, and Generic B have been thoroughly evaluated through bioequivalence studies in breeds like Holstein cattle. However, we emphasize the need to replicate these studies in Korean Hanwoo cattle due to their unique metabolic characteristics, which could significantly impact the efficacy and safety of drugs like ivermectin. Understanding these breed-specific responses is crucial for optimizing veterinary treatments and ensuring the well-being of these animals. We appreciate your insight and are committed to exploring this critical aspect in our future research.

Point 2: However, this project is not complete. Several critical parameters are missing. Despite the demonstrated bioequivalence, the efficacy of ivermectin by Korean Hanwoo cattle should be established or discussed.

Response 2: Thank you for your logical point. We have prioritized the bioequivalence assessment for products used in Korean cattle without pharmacokinetic data. Therefore, the bioequivalence assessment was conducted according to the procedure based on the formulation, analytical method and pharmacokinetic analysis. The next study will focus on the in vivo behavior of the pharmacokinetics according to the route of administration. In the next study, we will compare intravenous, intramuscular, and pour-on administration for a complete pharmacokinetic analysis.  Therefore, in this paper, we have added a section on analytical methods and comparative pharmacokinetics to the discussion in red.

“An important procedure required to perform pharmacokinetic studies of Ivermectin is the determination of the concentration of low levels of ivermectin in the blood after administration of the drug to cattle. The analysis of ivermectin in blood has been performed by HPLC. When performing ivermectin by HPLC, fluorescence analysis has been used after derivatization of ivermectin to measure low concentrations of the drug in blood. Kitzman et al (2006) used HPLC to analyze ivermectin in plasma after administration of ivermectin in humans, following fluorescence induction of ivermectin with trifluoroacetic anhydride (TFAA) and N-methylimidazole (NMI) [23]. The results showed that the limit of quantitation (LOQ) in human plasma for ivermectin was 0.2 ng/ml, which was more than five times the baseline noise observed for the retention time of ivermectin. The coefficient of variation (n=6) of the measured concentration at LOQ was 6.1%, and the deviation of the measured concentration from the mean and nominal value was 4.3%. However, this method is subject to losses during the fluorescence induction process, so the use of liquid chromatography-tandem mass spectrometry (LC-MS/MS), which extracts and analyzes the drug directly from plasma, may improve the accuracy of ivermectin analysis in plasma. The LOQ was 3 ng/ml, which was higher than the derivatization process of HPLC, but it was measurable in plasma after drug administration and higher than the MRL standard of 30 ug/ml. Therefore, the LC/MS/MS method is faster than the HPLC method, can prevent the loss of ivermectin caused by the fluorescence derivatization process, and can be analyzed accurately and quickly, and is expected to be useful for pharmacokinetics.”—

“Ivermectin has been the subject of many pharmacokinetic studies since its introduction in 1981 [28]. Depending on the species, oral, intramuscular (IM), subcutaneous (SC), or topical administration has been used. To evaluate the bioequivalence, the three formulations were administered topically at the same dose of 1 mg/kg to five Korean beef cattle in this study, each consisting of five animals. Blood was collected at a given time after topical administration and analyzed for plasma concentration by LC/MS instrumentation as shown in Fig 2. In cattle, the behavior of the drug in the body after intravenous and intramuscular injection of ivermectin showed a two-compartment model and a one-compartment mdoeld, respectively [29], [30]. However, Lanusse et al (1997) analyzed a two-compartment model [31]. Gayrard et al (1999) analyzed a one-compartment model behavior after pour on administration in cattle, which is consistent with the one-compartment behavior shown in this study [32]. Ivermectin has a Cmax of 0.022 μg/mL after intramuscular injection [30]. Subcutaneous administration, the most commonly used route of administration, reported a 6-fold difference in Cmax from a low of 0.022 μg/mL to a high of 0.133.2 μg/mL for the same formulation and for different formulations [13], [19], [30], [31]. This was thought to be due to differences in HPLC methods, differences in formulation and the route of administration. Gayrard et al (1999) reported a Cmax of 0.012 μg/mL and a Tmax of 81.6 hr after administration of ivermectin for the same topical application as in the present study [32]. Compared to the pharmacokinetic parameters obtained, the Cmax was 10 times higher and the Tmax was 2 times faster in this study. These differences were thought to be due to differences in HPLC and LC/MS analytical methods and loss of the drug due to the habit of licking in cattle experiments. This study was conducted in individual cages to prevent this. Therefore, in this study, a dose of 1 mg/kg was applied, which is twice as high as the previously used dose, considering the route of administration and previous Cmax results.”

Point 3 : Moreover, the faecal ivermectin detection results are not described as was pointed out in the text, line 410: the concentration-time profiles acquired from plasma and faecal samples from individual animals were subjected to analysis using the WinNonlin software.

Response 3 : Thank you for bringing to our attention the discrepancy regarding the faecal ivermectin detection results as mentioned in our manuscript, particularly in line 410. We acknowledge this oversight and regret any confusion it may have caused. We revised the sentence in line 410 to line 409.

The concentration-time profiles acquired from plasma in individual animals were subjected to analysis using the WinNonlin software (Version 6.1) developed by Statistical Consultants Inc. located in Lexington, KY.

Point 4: Why were the animals treated with a dose of 1 mg/kg of body weight?

Response 4: Thank you for your valuable feedback. The 1 mg/kg dose of ivermectin was used in clinical sector in Korea. We evaluated 1 mg/kg. When clinical trials were conducted by each company, 1mg/kg was found to be the most effective, so it was approved by the government and listed in the efficacy of the Korean government notice.

Point 5: The abstract is too extended and should be more concise for easy reading.

Response 5 : Thank you for your constructive feedback regarding the length and complexity of our abstract. We appreciate your suggestion to make it more concise and easier to read.

In our abstract, we now state the following:

“The objective of this study was to conduct a bioequivalence study of three formulations of ivermectin in Korean beef cattle after pour-on application at a dose of 1 mg/kg on the back of Korean native cattle (Hanwoo). To conduct bioequivalence testing, three groups (Control Innovator, Test Generic A and Test Generic B) of five clinically healthy Korean Hanwoo cattle (average weight 500 kg) were studied for pharmacokinetics. After topical administration to the skin, blood samples were drawn at given times. These blood samples were analyzed using a liquid chromatography-tandem mass spectrometry (LC-MS/MS) instrument. The time to reach maximum concentration (Tmax), maximum concentration (Cmax), and area under the curve (AUClast) of pharmacokinetic parameters were compared for bioequivalence. The results showed that the control had a Tmax of 41 ± 1.24 hr, a Cmax of 0.11 ± 0.01 μg/mL, and an AUClast of 9.33 ± 0 hr*μg/mL). The comparator Generic A had in a Tmax of 40 ± 1.14 hrs, a Cmax of 0.10 ± 0.01 ( μg/mL, and an AUClast of 9.41 ± 0.57 hr* μg/mL, while Generic B had in a Tmax of 40 ± 2.21 hr, a Cmax of 0.10 ± 0.01 μg/mL, and an AUClast of 9 hr* μg/mL. The bioequivalence indicators Cmax, Tmax and AUC values were all in the range of 80% to 120%, confirming that all three formulations tested were bioequivalent. In conclusion, the study showed that the two generic products were bioequivalent to the original patented product in the Hanwoo cattle.”

Point 6: Introduction: …there are few pharmacokinetic and clinical studies on the application of veterinary drugs in Korean cattle, particularly ivermectin…Line 78-82 The reason for the present bioequivalence study (innovator versus generics) (?) Please discuss.

Response 6 : Thank you for your insightful comments on our manuscript, particularly regarding the introduction and the rationale for our bioequivalence study of ivermectin in Korean cattle. There is no study on the pharmacokinetics and clinical efficacy of ivermectin products in Korean cattle. We described “The Hanwoo are primarily raised in South Korea, where their meat is highly prized. Therefore, drugs applied to the Hanwoo have been based on beef cattle. In particular, the unique physiological and metabolic characteristics of the Hanwoo cattle have not been well studied. Unlike the Holstein, the Hanwoo is a breed of small cattle native to Korea. In the past, they were used as raw livestock, but this has all but disappeared, and they are now primarily raised for their meat. The fur is brown and both males and females have horns. Maternal qualities are good, but milk production is low, less than 400 liters in a 170-day lactation [18].” at L71-78.

Point 7: Results: Concentration(ppm)–time (?) – line: 145; pour-on administration at 1000 g/kg.bw to cattle – line 153; Drug Equivalence Test Standards established by the Ministry of Food and Drug Safety – please reference – Line 189-190

Response 7: I appreciate your precise and detailed point. I'll make sure to correct all the mistakes on units (pour-on administration at 1 mg/kg.bw to cattle). We added the appropriate reference to this section to provide the necessary context. 33. “Lawrence, X.Y.; Li, B. V FDA Bioequivalence Standards; Springer, 2014; Vol. 13; ISBN 1493912526.”

Point 8: Discussion: Ivermectin is widely used in Korea as a topical pour-on method in Korean beef cattle for ectoparasite control, but there is no information on its pharmacokinetics- (lines 198-200). How were established the MRLs for ivermectin in the innovator?

Response 8 : Thank you for your query regarding the establishment of Maximum Residue Limits (MRLs). In the case of ivermectin used as a topical pour-on method in Korean beef cattle, the MRLs would have been established following similar comprehensive evaluations. In beef cattle, the MRL is based on the EMA's European public MRL assessment report (EPMAR). As with ivermectin (establishment of a maximum residue limit for muscle), the same level (MRL, 30 μg/kg) is applied to Korean Hanwoo cattle. As there are no reports of testing in Korean cattle, pharmacokinetic parameters were obtained for the three formulations and bioequivalence was confirmed.

Point 9 : The discussion should be more concise, considering the results and possible constraints  related to this study (e.g. Korean Hanwoo features concerning other breed cattle)

Response 9 : Thank you for your constructive feedback on the discussion section of our manuscript. We refined the discussion section to make it more focused and relevant to the study's specific context and limitations in the revised manuscript in red.

Reviewer 2 Report

Comments and Suggestions for Authors
  1. I am grateful for the opportunity to review the manuscript titled "Comparative Pharmacokinetics and Bioequivalence of Pour-On Ivermectin Formulations in Korean Hanwoo Cattle" for potential publication in Antibiotics. The manuscript is well-structured and possesses sufficient writing, content, novelty, and merit. However, I would like to bring attention to a few notable concerns that I believe require careful consideration and revision.
  2. The research title deserves commendation for its precision and informative approach. It effectively conveys the study's focus and offers readers a comprehensive understanding of its objectives.
  3. To ensure comparability, I suggest augmenting your detailed methodology description (4.1) by including the total volume of each formulation, along with the percentages or parts of each ingredient, if available. This supplementary information will provide a more comprehensive understanding of the composition of the formulations, facilitating a thorough evaluation of potential concentration variations between them.
  4. To enhance the data statistics and analysis section (4.7), I recommend verifying the normality of the data before proceeding with the analysis. This step would provide valuable insights and ensure the validity of the statistical tests conducted.
  5. Regarding the result described in section 2.1 ("Standard solution preparation"), I suggest reevaluating its relevance to the main findings and objectives of your study. While it provides information on the preparation of the stock and working solutions, it may not be crucial for readers to fully comprehend the significance of your research. If deemed non-essential, I recommend removing this section to streamline the manuscript and focus on the key results and conclusions.
  6. Please ensure that error bars are included for each data point in Figure 2. The incorporation of error bars will increase the clarity, reliability, and impact of the figures, allowing readers to make informed interpretations and potentially replicate the findings with greater confidence.
  7. Based on the results, which show no significant difference in absorption among the three formulations (as indicated by similar Tmax and Cmax values), it is important for the author to delve into the underlying factors contributing to the higher distribution (vd) and excretion (T1/2 and CL) observed specifically in the generic A formulation. By conducting a thorough analysis and providing an explanation for these findings, the author can greatly enhance the readers' understanding of the comparative pharmacokinetics of the three formulations. This comprehensive discussion will significantly contribute to the overall comprehension and interpretation of the study's results.
  8. To further enhance our understanding of the physiology and pharmacology in Korean Hanwoo cattle, I suggest you consider discussing the pharmacokinetics of ivermectin via pour-on medication in comparison to another cattle breed. This analysis would provide valuable insights into the potential similarities or differences in the pharmacokinetic profiles of ivermectin across different cattle breeds.

Author Response

Point 1 : I am grateful for the opportunity to review the manuscript titled "Comparative Pharmacokinetics and Bioequivalence of Pour-On Ivermectin Formulations in Korean Hanwoo Cattle" for potential publication in Antibiotics. The manuscript is well-structured and possesses sufficient writing, content, novelty, and merit. However, I would like to bring attention to a few notable concerns that I believe require careful consideration and revision.

Response 1 : Thank you for your thorough review and constructive feedback on our manuscript. We appreciate your recognition of the manuscript's structure, writing quality, content, novelty, and merit. We acknowledge the concerns you have raised and understand the importance of addressing them to enhance the quality of our work. We are committed to carefully considering each point and will undertake necessary revisions to address these issues comprehensively.

Point 2: The research title deserves commendation for its precision and informative approach. It effectively conveys the study's focus and offers readers a comprehensive understanding of its objectives.

Response 2 : Your acknowledgment of this effort is very encouraging. Thank you for your positive remarks.

Point 3 : To ensure comparability, I suggest augmenting your detailed methodology description (4.1) by including the total volume of each formulation, along with the percentages or parts of each ingredient, if available. This supplementary information will provide a more comprehensive understanding of the composition of the formulations, facilitating a thorough evaluation of potential concentration variations between them.

Response 3 : Thank you for your valuable suggestion regarding the methodology section of our manuscript. We will ensure to incorporate this data in our revised manuscript, thereby strengthening the comparability aspect of our research.

We have included the following improvements:

Ivermectin standard was provided by Sigma Aldrich (St. Louis, MO, USA). Acetonirile was purchased from Merck (Darmstadt, Germany). Formic acid was supplied by Fisher Scientific (Pittsburgh, PA, USA). All solvents used in the analysis were LC-MS grade. Purified water was obtained using a Milli-Q system (Millipore, Bedford, MA, USA).

Innovator (Formulation A, Group A): SY Himecin (Samyang Anipharm): Ingredients and amounts (out of 1L of the main preparation); Ivermectin (5g), Isopropanol(794.9mL), Propylene Glycol(8g), Isopropyl Myristate (160g), Oleyl Alcohol(32ml), Butylated Hydroxy Toluene (0.1g), Food Blue No.1 (Appropriate amount). Usage and capacity; Apply 0.1 mL of the main product (0.5 mg as Ivermectin) per kg of body weight as a single dermal application along the midline of the back.

Generic A (Formulation B, Group B): Gmectin-Pour On (GREEN CROSS Veterinary Products): Ingredients and amounts (out of 1mL of the main preparation); Ivermectin (5 mg), Triethanolamine (0.5mg), Food Blue No.1(0.01mg), Diethylene glycol monoethyl ether, Isopropyl myristate, Isopropanol (Appropriate amount). Usage and capacity; Dermal application of 1 mL of the product per 10 kg of body weight.

Generic B (Formulation C, Group C): IMEC-Pouron (SF company): Ingredients and amounts (out of 1L of the main preparation); Ivermectin (5 g), Propylene Glycol(8g), Isopropanol, Isopropyl Myristate, Food Blue No 1(Appropriate amount). Usage and capacity; Apply 0.1 mL of vehicle (0.5 mg as ivermectin) per kg of body weight as a single dermal application along the midline of the back.

Point 4: To enhance the data statistics and analysis section (4.7), I recommend verifying the normality of the data before proceeding with the analysis. This step would provide valuable insights and ensure the validity of the statistical tests conducted.

Response 4 : Thank you for your insightful suggestion regarding the data statistics and analysis section of our manuscript.

As you know, pharmacokinetic parameters are lognormally distributed. The lognormal distribution ranges from 0 to + infinity and is centered on the geometric mean of the population. The difference between the distributions is the limit on the left. In one case (lognormal), it is limited to zero. In the other case (normal), it is not bound at all.In real-world physiology, there are physiological limits to PK parameters. Therefore, PK parameters are lognormal because they are limited by physiological constraints rather than a normal distribution.

These limits require the use of different distribution assumptions. The methods used to characterize the pharmacokinetics (PK) and pharmacodynamics (PD) of compounds can be complex and sophisticated in nature.

 PK/PD analysis is a science that requires a mathematical and statistical background along with an understanding of biology, pharmacology, and physiology. Because PK/PD analysis guides critical decisions in drug development, such as optimizing dose, exposure frequency, and duration, it is critical to get these decisions right. Choosing the right analytical tool to make these decisions is critical. Fortunately, PK/PD analysis software has evolved significantly over the past few decades, allowing us to focus on a unified analysis. Therefore, we selected the Winnolin program (Phoenix, ver6.1, Cetera), which is widely used by the FDA and EUCAST.

Point 5: Regarding the result described in section 2.1 ("Standard solution preparation"), I suggest reevaluating its relevance to the main findings and objectives of your study. While it provides information on the preparation of the stock and working solutions, it may not be crucial for readers to fully comprehend the significance of your research. If deemed non-essential, I recommend removing this section to streamline the manuscript and focus on the key results and conclusions.

Response 5 : Your feedback regarding the relevance of the "Standard solution preparation" section in our manuscript is highly appreciated. Your recommendation to refine the manuscript for clarity and focus is invaluable, and we are committed to revising our work accordingly.

Point 6: Please ensure that error bars are included for each data point in Figure 2. The incorporation of error bars will increase the clarity, reliability, and impact of the figures, allowing readers to make informed interpretations and potentially replicate the findings with greater confidence.

Response 6 : Thank you for highlighting the importance of including error bars in Figure 2 of our manuscript. We are committed to implementing changes to enhance the quality of our presentation. We changed the Figure 2 in the revised manuscript.

Point 7: Based on the results, which show no significant difference in absorption among the three formulations (as indicated by similar Tmax and Cmax values), it is important for the author to delve into the underlying factors contributing to the higher distribution (vd) and excretion (T1/2 and CL) observed specifically in the generic A formulation. By conducting a thorough analysis and providing an explanation for these findings, the author can greatly enhance the readers' understanding of the comparative pharmacokinetics of the three formulations. This comprehensive discussion will significantly contribute to the overall comprehension and interpretation of the study's results.

Response 7: Thank you for your insightful observation regarding the results of our study on the comparative pharmacokinetics of three ivermectin formulations in Korean Hanwoo cattle.

The following description was added in the discussion part.

Ivermectin has been the subject of many pharmacokinetic studies since its introduction in 1981 [28]. Depending on the species, oral, intramuscular (IM), subcutaneous (SC), or topical administration has been used. To evaluate the bioequivalence, the three formulations were administered topically at the same dose of 1 mg/kg to five Korean beef cattle in this study, each consisting of five animals. Blood was collected at a given time after topical administration and analyzed for plasma concentration by LC/MS instrumentation as shown in Fig 2. In cattle, the behavior of the drug in the body after intravenous and intramuscular injection of ivermectin showed a two-compartment model and a one-compartment mdoeld, respectively [29], [30]. However, Lanusse et al (1997) analyzed a two-compartment model [31]. Gayrard et al (1999) analyzed a one-compartment model behavior after pour on administration in cattle, which is consistent with the one-compartment behavior shown in this study [32]. Ivermectin has a Cmax of 0.022 μg/mL after intramuscular injection [30]. Subcutaneous administration, the most commonly used route of administration, reported a 6-fold difference in Cmax from a low of 0.022 μg/mL to a high of 0.133.2 μg/mL for the same formulation and for different formulations [13], [19], [30], [31]. This was thought to be due to differences in HPLC methods, differences in formulation and the route of administration. Gayrard et al (1999) reported a Cmax of 0.012 μg/mL and a Tmax of 81.6 hr after administration of ivermectin for the same topical application as in the present study [32]. Compared to the pharmacokinetic parameters obtained, the Cmax was 10 times higher and the Tmax was 2 times faster in this study. These differences were thought to be due to differences in HPLC and LC/MS analytical methods and loss of the drug due to the habit of licking in cattle experiments. This study was conducted in individual cages to prevent this. Therefore, in this study, a dose of 1 mg/kg was applied, which is twice as high as the previously used dose, considering the route of administration and previous Cmax results.

Point 8: To further enhance our understanding of the physiology and pharmacology in Korean Hanwoo cattle, I suggest you consider discussing the pharmacokinetics of ivermectin via pour-on medication in comparison to another cattle breed. This analysis would provide valuable insights into the potential similarities or differences in the pharmacokinetic profiles of ivermectin across different cattle breeds.

Response 8 : Appreciating your insightful suggestion, I acknowledge the absence of prior data on the pharmacokinetics of pour-on ivermectin in Korean Hanwoo cattle. Your idea to contrast this with another cattle breed is certainly thought-provoking. One pharmacokinetic study was reported after application to the backs of beef cattle.

Gayrard et al (1999) analyzed a one-compartment model behavior after pour on administration in cattle, which is consistent with the one-compartment behavior shown in this study [32]. Ivermectin has a Cmax of 0.022 μg/mL after intramuscular injection [30]. Subcutaneous administration, the most commonly used route of administration, reported a 6-fold difference in Cmax from a low of 0.022 μg/mL to a high of 0.133.2 μg/mL for the same formulation and for different formulations [13], [19], [30], [31]. This was thought to be due to differences in HPLC methods, differences in formulation and the route of administration. Gayrard et al (1999) reported a Cmax of 0.012 μg/mL and a Tmax of 81.6 hr after administration of ivermectin for the same topical application as in the present study [32]. Compared to the pharmacokinetic parameters obtained, the Cmax was 10 times higher and the Tmax was 2 times faster in this study. These differences were thought to be due to differences in HPLC and LC/MS analytical methods and loss of the drug due to the habit of licking in cattle experiments. This study was conducted in individual cages to prevent this. Therefore, in this study, a dose of 1 mg/kg was applied, which is twice as high as the previously used dose, considering the route of administration and previous Cmax results.

Reviewer 3 Report

Comments and Suggestions for Authors

The aim of the study of the reviewed manuscript was to determine the pharmacokinetics and bioequivalence of three formulations of ivermectin used pour-on in Korean Hanwoo cattle.

Valuable elements of the manuscript include:

a) first comparative studies of three preparations of locally applied ivermectin in Korean Hanwoo cattle (no such data in the PubMed database so far)

b) application of the LC-MS/MS method for the determination of ivermectin in plasma.

However, the current version of the manuscript cannot be published due to its chaotic nature.

The second paragraph of the Introduction section should be devoted to a greater and more detailed extent to the results of studies on the pharmacokinetics of ivermectin in cattle.

The Results section contains content that should be in the Materials and Methods (e.g. subsection "2.1 Standard solution preparation" (lines 101-106); sentences: "To assess the validity ... were evaluated" (lines 122-123 ), "Linearity was assessed ... a standard solution" (lines 128-129), "The recovery and ... concentration level" (lines 131-132), "the sensitivity of ... ratio measurements (lines 134- 136).

The reference to Fig. 4 (line 109) is incorrect; reference should be made to Fig. 1.

The reference to Fig. 3 (line 113) is unclear.

The description of Fig. 2 should be more precise (number of animals in the group, drug dose, Hanwoo catle); in addition, the chart should contain means with standard deviation. The data presented in Table 2 are inconsistent with the data in the text (lines 173-177).

The description of Fig. 3 should be reworded. Currently, "for each" is used twice in the sentence (line 184). Moreover, "all the same" should be changed to similar (the results are not identical), ppm should be changed to μg and the description of the ordinate (Y axis) should be completed e.g. Concentration (μg/mL).

The "Discussion" subsection should be completely reworded because it has little relation to the data presented in Results section. First, the data regarding the analytical method should be exposed and compared with the methodologies of other authors. Then, the pharmacokinetics results (including bioequivalence) should be discussed and compared to analogous studies by other authors. Such a comparison is necessary because the assessed studies used a dose twice as high (1 mg/kg bw) compared to the available study results (0.5 mg/kg bw).

In the Materials and Methods section Merch (line 323) should be changed to Merck.

Reference to Fig. 1 (line 349) and should be to Fig. 4.

In Fig. 4 IVM Control should be changed to Innovator (this is in the figure description). The inclusion of the entry "Daily tracheal and rectal swab" in Fig. 4 is incomprehensible.

There is no information in subsections 4.4 and 4.5 about the range of the curve and the guidelines according to which the method was validated. In the case of an analytical method, at least the limit of detection, accuracy and precision should be given.

The entry "fecal samples" (line 410) should be deleted because there are no such results in the manuscript.

The description of the calculated parameters (lines 414-421) should include the same parameters that are presented in Results.

References should be verified because the current version contains vulnerabilities in positions 4, 23 and 26.

Author Response

Point 1 : The aim of the study of the reviewed manuscript was to determine the pharmacokinetics and bioequivalence of three formulations of ivermectin used pour-on in Korean Hanwoo cattle.
Valuable elements of the manuscript include:
a) first comparative studies of three preparations of locally applied ivermectin in Korean Hanwoo cattle (no such data in the PubMed database so far)
b) application of the LC-MS/MS method for the determination of ivermectin in plasma.
However, the current version of the manuscript cannot be published due to its chaotic nature.

Response 1 : Thank you for your feedback on our manuscript. We take your concerns about the manuscript's chaotic nature very seriously. It is clear that we need to significantly improve the organization and coherence of the manuscript to meet the publication standards. We entirely revised the manuscript in red.

Point 2: The second paragraph of the Introduction section should be devoted to a greater and more detailed extent to the results of studies on the pharmacokinetics of ivermectin in cattle.

Response 2 : Thank you for your valuable suggestion regarding the enhancement of the Introduction section of our manuscript in red. As you pointed out, we have made the following changes to the introduction and discussion in red.

[Introduction]

“The pharmacokinetics of ivermectin have been reported in a variety of animal species, including cattle, sheep, goats, pigs, horses, and dogs. Ali & Hennessy (1996) investigated the pharmacokinetic disposition and efficacy of ivermectin in sheep after administration of a feed mixture containing ivermectin [7]. Echeverria et al. (1997) reported the pharmacokinetics of ivermectin after intravenous and subcutaneous administration in cattle [8]. Gonzalez et al. (2006) conducted a study on the pharmacokinetics of a new formulation of ivermectin in goats [9]. Craven et al. (2001) reported the pharmacokinetics of moxidectin and ivermectin after intravenous injection in pigs with different body compositions [10]. Gokbulut et al. (2001) investigated the plasma pharmacokinetics and fecal excretion of ivermectin, doramectin, and moxidectin after oral administration in horses [11]. Daurio et al. (1992) reported the bioavailability of orally administered ivermectin in dogs [12]. Based on these reports, pharmacokinetic models indicate one or two compartments, depending on the route of administration and the formulation used. In addition, Lifschitz et al. (2004) performed a pharmacokinetic evaluation of four generic formulations in calves and found various pharmacokinetic parameters including Cmax, Tmax, and AUC in the aforementioned animal species [13]. These results suggest that the pharmacokinetic profile of ivermectin may be influenced by factors such as the body fat composition of the animal and the composition of the ivermectin formulation”.

[Discussion]

Ivermectin has been the subject of many pharmacokinetic studies since its introduction in 1981 [28]. Depending on the species, oral, intramuscular (IM), subcutaneous (SC), or topical administration has been used. To evaluate the bioequivalence, the three formulations were administered topically at the same dose of 1 mg/kg to five Korean beef cattle in this study, each consisting of five animals. Blood was collected at a given time after topical administration and analyzed for plasma concentration by LC/MS instrumentation as shown in Fig 2. In cattle, the behavior of the drug in the body after intravenous and intramuscular injection of ivermectin showed a two-compartment model and a one-compartment mdoeld, respectively [29], [30]. However, Lanusse et al (1997) analyzed a two-compartment model [31]. Gayrard et al (1999) analyzed a one-compartment model behavior after pour on administration in cattle, which is consistent with the one-compartment behavior shown in this study [32]. Ivermectin has a Cmax of 0.022 μg/mL after intramuscular injection [30]. Subcutaneous administration, the most commonly used route of administration, reported a 6-fold difference in Cmax from a low of 0.022 μg/mL to a high of 0.133.2 μg/mL for the same formulation and for different formulations [13], [19], [30], [31]. This was thought to be due to differences in HPLC methods, differences in formulation and the route of administration. Gayrard et al (1999) reported a Cmax of 0.012 μg/mL and a Tmax of 81.6 hr after administration of ivermectin for the same topical application as in the present study [32]. Compared to the pharmacokinetic parameters obtained, the Cmax was 10 times higher and the Tmax was 2 times faster in this study. These differences were thought to be due to differences in HPLC and LC/MS analytical methods and loss of the drug due to the habit of licking in cattle experiments. This study was conducted in individual cages to prevent this. Therefore, in this study, a dose of 1 mg/kg was applied, which is twice as high as the previously used dose, considering the route of administration and previous Cmax results.

Point 3 : The Results section contains content that should be in the Materials and Methods (e.g. subsection "2.1 Standard solution preparation" (lines 101-106); sentences: "To assess the validity ... were evaluated" (lines 122-123 ), "Linearity was assessed ... a standard solution" (lines 128-129), "The recovery and ... concentration level" (lines 131-132), "the sensitivity of ... ratio measurements (lines 134- 136).

Response 3 : Thank you for your insightful feedback regarding the organization of the manuscript. We acknowledge that certain content in the Results section, specifically the details about standard solution preparation and various methodological aspects, would be more appropriately placed in the Materials and Methods section.

Point 4: The reference to Fig. 4 (line 109) is incorrect; reference should be made to Fig. 1.

Response 4 : Thank you for pointing out the error in referencing Fig. 4 instead of Fig. 1 in our manuscript. We corrected this mistake in the manuscript to refer correctly to Fig. 1 at line 99.

Point 5: The reference to Fig. 3 (line 113) is unclear.

Response 5 : Thank you for pointing out the ambiguity surrounding the reference to Fig. 3 on line 113. Now we revised the line 113.

The mass-to-charge ratio (m/z) and retention time of ivermectin from the spiked plasma samples were similar to those obtained from the standard solutions (Fig. 2).

Point 6: The description of Fig. 2 should be more precise (number of animals in the group, drug dose, Hanwoo catle); in addition, the chart should contain means with standard deviation. The data presented in Table 2 are inconsistent with the data in the text (lines 173-177).

Response 6 : Thank you for your valuable feedback. We revised the line 166-171.

Point 7: The description of Fig. 3 should be reworded. Currently, "for each" is used twice in the sentence (line 184). Moreover, "all the same" should be changed to similar (the results are not identical), ppm should be changed to μg and the description of the ordinate (Y axis) should be completed e.g. Concentration (μg/mL).

Response 7: Thank you for your valuable feedback on the description of Fig. 3. We revised the legends as follows : Comparing the maximum concentration (Cmax) and the area under the curve to the last measurable concentration (AUClast) for the control Innovator, Generic A, and Generic B ivermectin formulations in cattle. The values for Cmax and AUClast for each of the three products were identical, showing no statistically significant variations. We changed the Figures in revised manuscript.

Point 8: The "Discussion" subsection should be completely reworded because it has little relation to the data presented in Results section. First, the data regarding the analytical method should be exposed and compared with the methodologies of other authors. Then, the pharmacokinetics results (including bioequivalence) should be discussed and compared to analogous studies by other authors. Such a comparison is necessary because the assessed studies used a dose twice as high (1 mg/kg bw) compared to the available study results (0.5 mg/kg bw).

Response 8 : Thanks for pointing that out, I've revised, added, and organized that subset of the discussion again in red.

Point 9 : In the Materials and Methods section Merch should be changed to Merck (line 307).

Response 9 : Thank you for pointing out the typographical error in the Materials and Methods section. We provide the revised version.

Point 10 : Reference to Fig. 1 (line 361) and should be to Fig. 4.

Response 10 : Thank you for highlighting the erroneous reference to Fig. 1 on line 342. We corrected the sentence.

Point 11 : In Fig. 4 IVM Control should be changed to Innovator (this is in the figure description). The inclusion of the entry "Daily tracheal and rectal swab" in Fig. 4 is incomprehensible.

Response 11 : Thank you for your observations regarding Fig. 4. We acknowledge the need to correct the labeling in the figure description. We reviewed and corrected the Figure 4.

Point 12 : There is no information in subsections 4.4 and 4.5 about the range of the curve and the guidelines according to which the method was validated. In the case of an analytical method, at least the limit of detection, accuracy and precision should be given.

Response 12 : Thank you for highlighting the omission of key details regarding the range of the curve and the validation guidelines for the analytical method in subsections 4.4 and 4.5. We ensured that essential data such as the limit of detection, accuracy, and precision are clearly presented.

In this study, we validated a novel analytical method by spiking blank samples, adhering to the CD 2002/657/EC guidelines for method validation. The validation process encompassed a thorough evaluation of several key parameters, including the linearity of the calibration curve, specificity, limit of detection (LOD), limit of quantification (LOQ), accuracy, and precision. Calibration was meticulously conducted over a range of 10 to 200 µg/kg to ensure the method's applicability across various concentrations. The LOD and LOQ were carefully determined from the lowest detectable concentration of the analyte, demonstrating signal-to-noise (S/N) ratios of 3 and 10, respectively, which underscore the method's sensitivity. Additionally, the method's specificity was confirmed through the analysis of both solvent blanks and spiked samples, with a particular focus on the absence of background interference at the retention times of the target compounds. This absence of interference is a strong indicator of the method's reliability and effectiveness in differentiating the target compounds from potential contaminants.

Point 13 : The entry "fecal samples" (line 410) should be deleted because there are no such results in the manuscript.

Response 13 : Thank you for bringing to our attention the discrepancy regarding the faecal ivermectin detection results as mentioned in our manuscript, particularly in line 410. We acknowledge this oversight and regret any confusion it may have caused. We revised the sentence in line 410. The concentration-time profiles acquired from plasma in individual animals were subjected to analysis using the WinNonlin software (Version 6.1) developed by Statistical Consultants Inc. located in Lexington, KY.

Point 14 : The description of the calculated parameters (lines 414-421) should include the same parameters that are presented in Results.

Response 14 : Thank you for pointing out the discrepancy between the parameters described in the Materials and Methods section . We revised the line (409-418) as follows :The use of NCA allowed for the determination of key pharmacokinetic parameters. In addition, the analysis included the determination of the maximum plasmatic ivermectin concentration (Cmax) and the corresponding time required to reach this peak value (Tmax). The estimation of the terminal-phase disposition rate con-stant (λ) was accomplished through a least-squares linear regression of the natural log-transformed ivermectin concentrations over time. The half-life (T1/2) was subsequently calculated using the formula t1/2 = 0.693/λ.

Point 15 : References should be verified because the current version contains vulnerabilities in positions 4, 23 and 26.

Response 15 : Thanks for pointing out that the reference was not formatted correctly. We've fixed it for the Antibiotics version of MDPI.

Round 2

Reviewer 1 Report

Comments and Suggestions for Authors

I thank the authors for their answers to the questions asked, which were correctly formulated. However, there are some points in the text that I propose to erase:

Line 29: the original patented product in the Hanwoo cattle. Please, erase the word patented.

Line 54-55: Daurio et al. (1992) reported the bioavailability of orally administered ivermectin in dogs [12].Please erase this sentence.

Author Response

Comment 1. I thank the authors for their answers to the questions asked, which were correctly formulated. However, there are some points in the text that I propose to erase:

Line 29: the original patented product in the Hanwoo cattle. Please, erase the word patented.

Line 54-55: Daurio et al. (1992) reported the bioavailability of orally administered ivermectin in dogs [12]. Please erase this sentence.

Response 1. Thank you for your comments. Your comment seems to organize the paper. We have removed all of these sentences as you pointed out. 

Reviewer 2 Report

Comments and Suggestions for Authors

Thank you for your response and for addressing the concerns raised in my review. I appreciate the improvements you have made, particularly in the methodology section (4.1 and 4.7) and the inclusion of error bars in Figure 2. I also commend your efforts to provide a comprehensive discussion on the comparative pharmacokinetics of the three formulations for addressing the Cmax and Tmax values (#point 6). However, there are no statistically significant differences regarding the AUC parameter. To further enhance the readers' understanding of the comparative pharmacokinetics of the three formulations, it would be beneficial if you could also discuss the underlying factors contributing to the shorter T1/2 and lower Vd observed specifically in the generic A formulation, as compared to the Innovator and generic B formulations. Overall, I accept the manuscript in its present form.

Author Response

Comment 1. Thank you for your response and for addressing the concerns raised in my review. I appreciate the improvements you have made, particularly in the methodology section (4.1 and 4.7) and the inclusion of error bars in Figure 2. I also commend your efforts to provide a comprehensive discussion on the comparative pharmacokinetics of the three formulations for addressing the Cmax and Tmax values (#point 6). However, there are no statistically significant differences regarding the AUC parameter. To further enhance the readers' understanding of the comparative pharmacokinetics of the three formulations, it would be beneficial if you could also discuss the underlying factors contributing to the shorter T1/2 and lower Vd observed specifically in the generic A formulation, as compared to the Innovator and generic B formulations. Overall, I accept the manuscript in its present form.

Response 2 : Thank you for your comments. As you pointed out, it is formulations used in Korean Hanwoo. The difference is the presence or absence of Propylene glycol. And your comment can be a guide when developing a new formulation. We have added the following to the discussion based on your comment;
“In this study, no statistical differences were seen when comparing the pharmacokinetic parameters of the three formulations; however, the Innovator and Generic B formulations tend to exhibit longer T1/2 and higher Vz/F than those observed in Generic A formulation. These differences were likely due to the fact that the Innovator and Generic B formulations contained Propylene Glycol at the same concentration, while Generic A formulation excluded this ingredient. From these results, it was inferred that Propylene Glycol may have an effect on the critical T1/2 and Vz/F when applied topically (pour-on. All three formulations used in the present study were formulated in various fat-soluble solvents with or without propylene glycol. After topical application of these formulations at 1 mg/kg, all of them showed Cmax of 0.1-0.11 μg/mL, Tmax of 40 hr, and AUClast of 9.33-9.75 hr*μg/mL and were found to be bioequivalent. However, the effect on propylene glycol is not statistically significant in the study. Despite of the result, the study of formulation would be an urgent and important topic for further research.” We highlighted in yellow the revised parts.